# Effect of Biochar in Modulating Anaerobic Digestion Performance and Microbial Structure Community of Different Inoculum Sources

**Jingran Ding [1,2,†], Feng Zhen [2] , Xiaoying Kong [2], Yunzi Hu [2], Yi Zhang [2,3,*] and Lang Gong [1,4,*]**

1   College of Veterinary Medicine, South China Agricultural University, Guangzhou 510462, China; jingran@scau.edu.cn
2   Guangzhou Institute of Energy Conversion, Chinese Academy of Sciences, Guangzhou 510640, China; zhenfeng@ms.giec.ac.cn (F.Z.); kongxy@ms.giec.ac.cn (X.K.); huyunzi@ms.giec.ac.cn (Y.H.)
3   Institute for Polymer Synthesis, Kingfa Sci. & Tech. Co., Ltd., Guangzhou 510663, China
4   Maoming Branch, Guangdong Laboratory for Lingnan Modern Agriculture, Maoming 525000, China
*   Correspondence: zhangyi@ms.giec.ac.cn (Y.Z.); gonglang@scau.edu.cn (L.G.)
†   The first author of this paper.

**Abstract:** Biochar has attracted increasing attention as an additive for enhancing the performance of anaerobic digestion (AD), but the effect of biochar on microbial regulatory mechanisms in enhancing AD performance is unclear. To investigate how biochar modulates the process of AD, different inoculum sources including cellulose–peptone–swine inoculum (CPI) and swine manure inoculum (SMI) were designed to determine the effect of biochar on the performance and microbial communities of anaerobic digestion of the feedstock concentration from 1 to 6%. The results showed that the methane yields of CPI seeds were higher 20.3–38.7% than those of SMI seeds without a biochar addition, whereas the biochar addition reduced 5.3 and 23.1% of the corresponding methane yield of CPI and SMI, respectively. The biochar enhances the accumulation of volatile fatty acids (VFAs) and weakens the potential ammonia inhibition by adsorption, and it can improve the degradation rate of organic content of soluble COD for different inoculum sources. Microbial community analyses showed that the biochar addition could facilitate the growth of *Bacteroidetes* and *Clostridiales*, and it enriched the relative abundance of hydrogenotrophic methanogens *Methanobrevibacter* and *Methanobacterium*. Overall, although the modulation of biochar possessed different effects on the anaerobic digestion performance, it contributed to the stability and degradation efficiency of the digestion system. The recycling implication of biochar is critical to realizing a low-carbon and renewable treatment system for organic wastes.

**Keywords:** anaerobic digestion; biochar; modulate; different inoculum; microbial structure community



## 1. Introduction

Increasing human demand for meat and animal products has boosted the development of livestock farming, which has consequently generated a large amount of manure. In China, over 600 million tons of swine manure are annually produced by farms [1]. Improper disposal of swine manure that contains abundant nutrients and organic substances can cause serious environmental pollution, such as contamination of ground and surface water and transmission of pathogenic microorganisms [2]. Anaerobic digestion (AD) is an environmentally friendly, efficient, and low-carbon-emission method for treating livestock and poultry manure [3]. AD not only allows degraded swine manure to be used as an agricultural fertilizer but also produces clean energy biogas that is fed into the natural gas network [4]. Thus, AD has tremendous application potential.

Anaerobic digestion is a multistage biochemical process that includes hydrolysis, acidification, and methanogenesis and is a dynamic collaborative process maintained by

microorganisms cooperating and restricting with each other. However, the high concentrations of ammonia and volatile fatty acids (VFAs) attributed to the decomposition of proteins and urea in swine manure could affect anaerobic microorganisms responsible for the biological conversion of organic material into methane, which ultimately reduces methane production [5]. Methanogenic activities can be inhibited with $NH_4^+$-N concentrations >1700 mg/L [6]; acetotrophic methanogens in AD are more vulnerable than hydrogenotrophic methanogens under ammonia stress [7]. Meanwhile, high ammonia levels can alter microbial community structure, ultimately affecting AD performance [8]. Therefore, improving the ability of microorganisms to thrive in harsh environments and degrade VFAs or $NH_4^+$-N during AD is crucial. However, maintaining a balance in the transformation between substrates and microbes through the optimization of conventional process parameters, such as pH and temperature, remains challenging. Domesticating microorganisms and additives are effective strategies to adjust the microbial transformation process during AD. Jo et al. [9] investigated ammonia toxicity and acclimation of anaerobic microbiomes in the continuous AD of swine manure and found that methane production significantly decreased at a total ammonia nitrogen concentration of 2.5 g/L, whereas colony domestication could mitigate ammonia inhibition of methane production. Moreover, additives, such as nanobubble water [10], zero-valent iron [11], and biochar [12], can improve the digestion performance of AD. Indeed, carbon materials have shown the most promising results.

Biochar, a carbon material prepared from agricultural waste via pyrolysis or hydrothermal methods, can be widely used in many fields due to its favorable physical and chemical properties. For instance, difficult-to-degrade raw materials after AD or other agricultural biomass wastes can be converted into biochar via pyrolysis, and storing the resulting biochar in soils has great potential to improve the quality of soils and carbon sequestration. In addition, biochar can be mixed with conventional fertilizers, such as compound fertilizers containing nitrogen, phosphorus, and potassium, urea fertilizers, poultry manure, compost, and organic fertilizers, prior to or during crop cultivation, and is effective in alleviating the constraints of problem soils on plant growth or crop production, thus increasing crop yield [13]. Biochar also can be used in the animal farming industry and as a feed additive for animals showing benefits in terms of animal growth, gut microbiota, reduced enteric methane production, egg yield, and endo-toxicant mitigation [14]. Furthermore, owing to the advantages of thermochemical stability, excellent thermal and electrical conductivity and adsorption, large specific surface area, strong ion-exchange properties, high porosity, and easy regeneration, it is used to absorb pollutants from water [15]. At present, the use of biochar materials to regulate AD processes has attracted interest due to the structural and electrochemical properties of biochar. For example, VFA generation in the acidification stage and degradation during methanogenesis is improved by biochar supplementation in an AD system [16]. Lü et al. [17] demonstrated that biochar can alleviate the accumulation of organic acids and shorten the lag phase. Shen et al. [18] used corn stover biochar for AD and showed that the removal rate of $CO_2$ was as high as 86.3%, while the average methane content increased by 42.4%. Wang et al. [19] added biochar into AD, finding that maximum $CH_4$ production rate was increased. Biochar derived from dairy manure increases methane yield by up to 35.7% at a dosage of 10 g/L [20]. Using biochar in a digestion system can improve process stability and methane yield, but the inhibition caused by that should be considered before using biochar. A study has shown that the addition of 5% biochar did not significantly affect biogas production compared with 0% biochar in AD, whereas the addition of 20–50% biochar decreased biogas production [21].

From the above brief review, it can be seen that many researchers have focused on the methane production effect of biochar on anaerobic fermentation and its related mechanisms, and fewer studies have focused on the relationship between biochar and the performance of anaerobic fermentation containing different inoculants. Therefore, to illustrate the effect of biochar addition on anaerobic digestion performance with different inoculum sources, this study aimed to investigate the biogas production performance and microbial

community structure under different substrate concentrations during the entire stage of AD. In addition, we analyzed the prospects and implications of biochar application in future anaerobic digestion systems.

## 2. Materials and Methods

### 2.1. Swine Manure and Inoculum

Swine manure was collected from Desheng Modern Farm in Guangzhou City, the Guangdong Province, China, and swine manure inoculum (SMI) was obtained from an anaerobic digester, operated as a 50 L mesophilic (37 ± 0.5 °C) continuous stirred-tank reactor (CSTR) in the Laboratory of Biomass Biochemical Conversion, Guangzhou Institute of Energy Conversion (GIEC), Chinese Academy of Science (CAS), and fed with fresh swine manure. The cellulose–peptone–swine inoculum (CPI) was obtained from a 50 L laboratory thermophilic (37 ± 0.5 °C) CSTR and enriched by the addition of *α-cellulose*, bacterial peptone, and fresh swine manure. The enrichment cycle lasted for 7 d. Within day 4, *α-cellulose* and bacterial peptone were mixed at a ratio of 2:1 and added to the CSTR at a concentration of 1 g/L. Feeding was stopped on days 5 and 7, and swine manure was added to the CSTR at 1 g/L on day 6. The enrichment process consisted of eight cycles. Prior to use, the inoculum was sieved through a 1 mm mesh to remove grit and other solids. Coconut shell biochar was obtained from McLean Ltd. (Guangzhou, China). It was prepared from the waste of coconut shells, which were subjected to anaerobic conditions at 450 °C and, thereafter, were crushed and screened with 40 to 100 mesh prior to use. Biochar of different particle sizes was mixed before conducting the experiment. The characteristics of the feedstock, CPI, SMI, and coconut shell biochar are listed in Table 1.

**Table 1.** Characteristics of feedstock, pig manure inoculum, laboratory inoculum, and coconut shell biochar.

| Parameter | TS (%FM) | VS (%FM) | VS/TS (%) | C (%TS) | H (%TS) | N (%TS) | C/N (%TS) | BET Surface Area (m$^2$/g) | Pore Diameter (nm) |
|---|---|---|---|---|---|---|---|---|---|
| Feedstock | 33.48 ± 0.01 | 26.25 ± 0.01 | 78.41 ± 0.003 | 39.75 ± 0.6 | 5.95 ± 0.09 | 2.64 ± 0.4 | 14.28 ± 0.4 | | |
| Cellulose–peptone–swine inoculum | 1.12 ± 0.1 | 0.45 ± 0.1 | 41.61 ± 0.005 | | | | | | |
| Swine manure inoculum | 2.29 ± 0.1 | 1.44 ± 0.1 | 62.83 ± 0.002 | | | | | | |
| Coconut shell biochar | | | | 77.68 | 1.57 | 0.695 | 111.76 | 134.47 | 4.80 |

Data are presented as mean ± standard deviation (n = 3). TS: total solids; FM: fresh matter; and VS: volatile solids.

### 2.2. Experimental Setup for the AD System

Batch anaerobic digestion experiments were performed in a 500 mL reactor with a working volume of 400 mL under mesophilic conditions controlled at 37 ± 0.5 °C. Eight experimental groups were established, within six of which CPI and SMI were added to the swine manure with total solid (TS) concentrations of 1, 3, and 6%, and labeled as CPI1, CPI3, CPI6, SMI1, SMI3, and SMI6, respectively, whereas biochar was added to swine manure at a TS concentration of 3% with CPI and SMI, labeled as BCPI3 and BSMI3 to the remaining two groups at a biochar: swine manure ration of 1:1. After the feedstock, inoculum and biochar were added to reactors, all the reactors were flushed with nitrogen gas for 1 min in the headspace, and sealed with rubber stoppers to maintain anaerobic state. All experimental groups were set up separately with three replicates; the blanks containing the same amount of CPI and SMI were set up in parallel. The AD experiment was conducted for 30 d.

### 2.3. Analytical Methods

Total solids (TS) and volatile solid (VS) levels were measured according to standard methods [22]. The elemental mass fractions of C, H, and N were determined using a Vario EL elemental analyzer. The concentrations of NH$_3$-N and soluble chemical oxygen demand

(SCOD) were determined using a commercially available kit and a spectrophotometer according to the manufacturer's instructions (Hach, Loveland, CO, USA) [23,24]. Electrical conductivity was measured using a portable conductivity meter (CON200, Guangzhou, China). The gas composition was determined using a gas chromatograph (GC-2014, Shimadzu, Kyoto, Japan). VFA concentration was determined using an HPLC system (Model e2698, Waters, San Diego, CA, USA) equipped with a Bio-RAD column at 50 °C and 0.005 M $H_2SO_4$ as the mobile phase at a flow rate of 0.5 mL/min. The pH was measured using an FE28-Standard meter (Mettler-Toledo, Zurich, Switzerland) calibrated with standard buffer solutions of pH 4.0, 7.0, and 10.0 before testing.

### 2.4. Microbial Community Analysis

To analyze the bacterial and archaeal community, samples were obtained on days 1, 5, and 10 and stored at $-20$ °C for further analysis. DNA was extracted using an E.Z.N.A$^{TM}$ Mag-Bind Soil DNA Kit (Omega, M5635-02, Norcross, GA, USA), following the manufacturer's instructions, while Qubit 4.0 (Thermo, Waltham, MA, USA) was used to measure the concentration of DNA to ensure that an adequate amount of high-quality genomic DNA was extracted. The PCR forward primer 341F (5′-CCTACGGGNGGCWGCAG-3′) and PCR reverse primer 805R (5′-GACTACHVGGGTATCTAATCC-3′) were selected for amplifying the V3-V4 hypervariable regions of bacterial 16S rRNA gene using 2×Hieff$^{®}$ Robust PCR Master Mix (Yeasen, 10105ES03, Shanghai, China). Hieff NGS$^{TM}$ DNA Selection Beads (Yeasen, 10105ES03, Shanghai, China) were used to purify the free primers and primer dimer species in the amplicon product. Samples were delivered to Sangon BioTech (Shanghai, China) for library construction using a universal Illumina adaptor and index. Sequencing was performed using the Illumina MiSeq system (Illumina MiSeq, San Diego, CA, USA), according to the manufacturer's instructions. After sequencing, the two short Illumina readings were assembled by PEAR software (version 0.9.8) according to the overlap, and fastq files were processed to generate individual fasta and qual files, which could then be analyzed by standard methods. The effective tags were clustered into operational taxonomic units (OTUs) of ≥97% similarity using Usearch software (version 11.0.667). Chimeric sequences and singleton OTUs (with only one read) were removed, after which the remaining sequences were sorted into each sample based on the OTUs. The tag sequence with the highest abundance was selected as a representative sequence within each cluster. Bacterial representative sequences were classified taxonomically by blasting against the RDP Database.

### 2.5. Statistical Analysis

All experiments were carried out in triplicate, and the results were expressed as mean ± SE. All statistical analyses were performed using SPSS version 23.0. Analysis of variance (ANOVA) was used to detect the significant influence of SMI, CMI, and the addition of biochar on the concentration of $NH_4^+$-N and SCOD, the degradation rate of SCOD, and Electrical conductivity in AD. Differences in means were determined using Tukey's test, and significant differences were set at $p < 0.05$.

Statistical software R with a Vegan package was used to perform the operational taxonomic unit (OTU) and phylotype-based analyses of both bacterial and archaeal communities. The $\alpha$-diversity indices (including Chao, Simpson, ACE, and Shannon indices) were quantified in terms of OTU richness. The sampling coverage was calculated based on Good's methods. To assess sample adequacy, all $\alpha$ diversity indices were calculated with Mothur software (version 3.8.31). To estimate the impact of biochar on the diversity of the microbial community of the sample, we calculated the within-sample (alpha) diversity by T test for two groups, and multiple group comparisons were made using the ANOVA test.

## 3. Results and Discussion

### 3.1. Anaerobic Digestion Performance

3.1.1. Variations in $NH_4^+$-N, SCOD, and Electrical Conductivity (EC)

Excessive amounts of nitrogen are produced during the AD of swine manure, leading to the accumulation of $NH_4^+$-N. Ammonia nitrogen concentration in all experimental groups was higher than 2300 mg/L, and in the SMI groups, it was all higher than that in the CPI groups (Figure 1). Compared with day 1, the concentration of $NH_4^+$-N on day 30 increased in experimental groups without the addition of biochar, while it was decreased in experimental groups with the supplement of biochar. It is worth noting that the concentration of $NH_4^+$-N on day 30 was 11.39% ($p < 0.05$) lower than that on day 1 in BCPI3, while it in BSMI3 was 14.95% ($p < 0.01$) lower than that on day 1. A possible explanation is that the high protein and urea content in swine manure was degraded into $NH_4^+$-N during AD, which increased $NH_4^+$-N concentration, while the resulting concentration of $NH_4^+$-N decreased due to the absorption of biochar. A previous study also demonstrated that the biochar had an adsorption ability of $NH_4^+$-N, which led to a decrease in $NH_4^+$-N concentration in anaerobic digestion [25]. The concentration of SCOD in all experimental groups on day 30 was significantly reduced compared to that on day 1. The SCOD removal efficiency of the groups without biochar was all below 50%, but the SCOD removal efficiency of BSMI3 was 68.06 ± 0.09%, which was significantly higher than that of SMI3 by 58.7%. Moreover, the degradation rate of SCOD in BCPI3 was also improved compared with that in CPI3, which was 6.09% higher than that in CPI3. This may be attributed to the accumulated ammonia nitrogen being a risk factor threatening the stability of the AD system and lowering its capacity to deal with SCOD [26]. However, as biochar adsorbs organic matter and creates a surface area for the colonization of microbial cells [27], adding biochar helped the AD system resist the negative effects of ammonia nitrogen, facilitated the degradation of organic matter, and increased the SCOD removal efficiency.

**(a)**
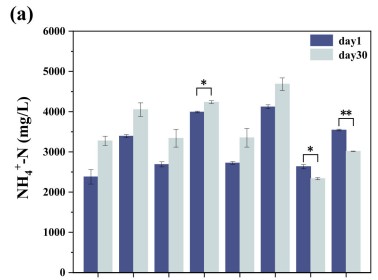

**(b)**
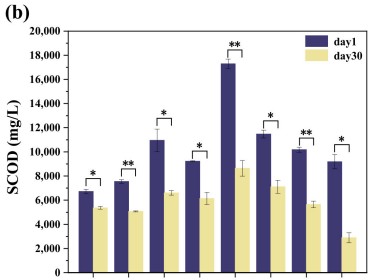

**(c)**
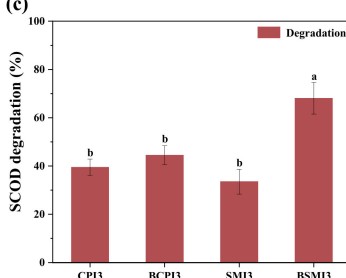

**Figure 1.** Profiles of (**a**) $NH_4^+$-N concentration of different TS concentrations of SM and adding biochar to CPI and SMI on day 1 and day 30. (**b**) SCOD concentration of different TS concentrations of SM and adding biochar to CPI and SMI on day 1 and day 30. (**c**) Degradation of SCOD concentration by adding biochar to CPI and SMI. Asterisk denotes statistically significant differences * $p < 0.05$; ** $p < 0.01$. Different letters indicate significant differences, $p < 0.05$.

Table 2 shows the variation of EC in AD. Regardless of whether on day 1 or 30, EC in SMIs was higher than CPIs, suggesting that different inocula have different electrical conductivity. A possible reason for this could be that SMI consumes more energy to synthesize the naturally conductive pili necessary for DIET to adjust to environmental conditions [28], which increases EC. Additionally, compared with CPI3 and SMI3, EC in BCPI3 and BSMI3 was higher owing to the addition of biochar, which allows for electron transfer via the conductance of carbon matrices and promotes DIET and efficient electron transfer [29]. The application of highly conductive and biocompatible conductive polymers can significantly improve biological electron transfer kinetics in various microbial electrochemical systems [30] and possibly facilitate DIET between bacteria and methanogens, thus accelerat-

ing the conversion of various reduced organic compounds in AD [29], which corresponds to higher degradation of SCOD and reduction of ammonia nitrogen concentration in BSMI3.

**Table 2.** Variations of EC in different groups during AD.

| Group | EC (ms/cm) | | | | | |
|---|---|---|---|---|---|---|
| | **1 d** | **3 d** | **5 d** | **10 d** | **20 d** | **30 d** |
| CPI1 | 17.03 ± 0.07 [abc] | 17.20 ± 0.02 [b] | 17.29 ± 0.12 [ab] | 17.01 ± 0.11 [bc] | 17.15 ± 0.07 [abc] | 16.60 ± 0.03 [a] |
| SMI1 | 20.23 ± 0.24 [d] | 19.93 ± 0.17 [d] | 18.16 ± 0.12 [b] | 21.29 ± 0.03 [f] | 18.11 ± 0.22 [d] | 19.21 ± 0.12 [c] |
| CPI3 | 16.66 ± 0.06 [ab] | 16.72 ± 0.03 [ab] | 16.81 ± 0.16 [ab] | 16.36 ± 0.24 [ab] | 16.92 ± 0.04 [ab] | 16.64 ± 0.02 [a] |
| SMI3 | 19.22 ± 0.25 [d] | 19.02 ± 0.14 [cd] | 17.56 ± 0.06 [ab] | 20.74 ± 0.13 [f] | 18.13 ± 0.15 [d] | 19.03 ± 0.01 [bc] |
| CPI6 | 15.49 ± 0.03 [a] | 15.49 ± 0.30 [a] | 15.93 ± 0.24 [a] | 15.71 ± 0.03 [a] | 16.47 ± 0.07 [a] | 16.44 ± 0.03 [a] |
| SMI6 | 18.70 ± 0.06 [cd] | 17.66 ± 0.65 [bc] | 16.95 ± 0.88 [ab] | 19.94 ± 0.11 [e] | 17.84 ± 0.19 [cd] | 18.74 ± 0.08 [b] |
| BCPI3 | 17.94 ± 0.03 [bc] | 17.49 ± 0.09 [b] | 16.28 ± 0.12 [ab] | 17.89 ± 0.09 [d] | 17.40 ± 0.03 [bc] | 18.71 ± 0.13 [b] |
| BSMI3 | 16.69 ± 0.53 [ab] | 17.65 ± 0.14 [bc] | 16.98 ± 0.26 [ab] | 17.67 ± 0.1 [cd] | 18.23 ± 0.01 [d] | 19.70 ± 0.01 [d] |

Different letters indicate significant differences, $p < 0.05$.

In conclusion, the addition of biochar to AD can improve the degradation of organics and removal of SCOD by resisting the negative effects of ammonia nitrogen and enhancing electron transfer.

### 3.1.2. Biogas Production Performance

The cumulative methane yield and daily methane production from the AD of swine manure with different TS concentrations over 30 d are illustrated in Figure 2. Methane production performance in the experimental groups with added enriched cellulose–peptone–swine inoculant was superior to that added non-enriched swine manure inoculant groups. For the cumulative methane yield, CPI1, CPI3, and CPI6 achieved 144.33, 151.29, and 149.97 mL/g TS, respectively, corresponding to 34.7, 38.7, and 20.3% increases compared with SMI1, SMI3. And SMI6. The initial increase in daily methane production was attributed to the fast conversion of readily digestible fractions, such as monosaccharides and proteins [31]. As shown in Figure 2b, CPI1, CPI3, and CPI6 exhibited similar trends, with their first peak times occurring on day 1, earlier than those of SMI1, SMI3, and SMI6, which can be explained by the faster organic degradation achieved in the CPI groups. Moreover, the addition of biochar not only did not improve methane production performance but had an inhibitory effect. In Figure 2c,d, the cumulative methane yield in BCPI3 was 143.4 mg/L TS, while that in BSMI3 was 71.32 mg/L TS, which was 5.3 and 23.1% lower than that in CPI3 and SMI3, respectively. The promotion of methane production in BCPI3 appeared between days 3 and 8; subsequently, it had been in a stage of inhibition of methane production, whereas methane production in BSMI3 was in a stage of inhibition from the second day during AD. Therefore, adding biochar had a more severe inhibitory effect on SMI than CPI groups, which corresponds to the previous variation in EC related to the oxidation–reduction potential. Although the functional groups in biochar provide various redox properties, the experimental results showed that the biochar used in this study had few redox-active organic functional groups, thus leading to a high potential that can negatively affect methanogens [32]. Wang et al. [33] revealed that biochar with higher EC but lower redox-active properties did not significantly enhance the methane production rate [28].

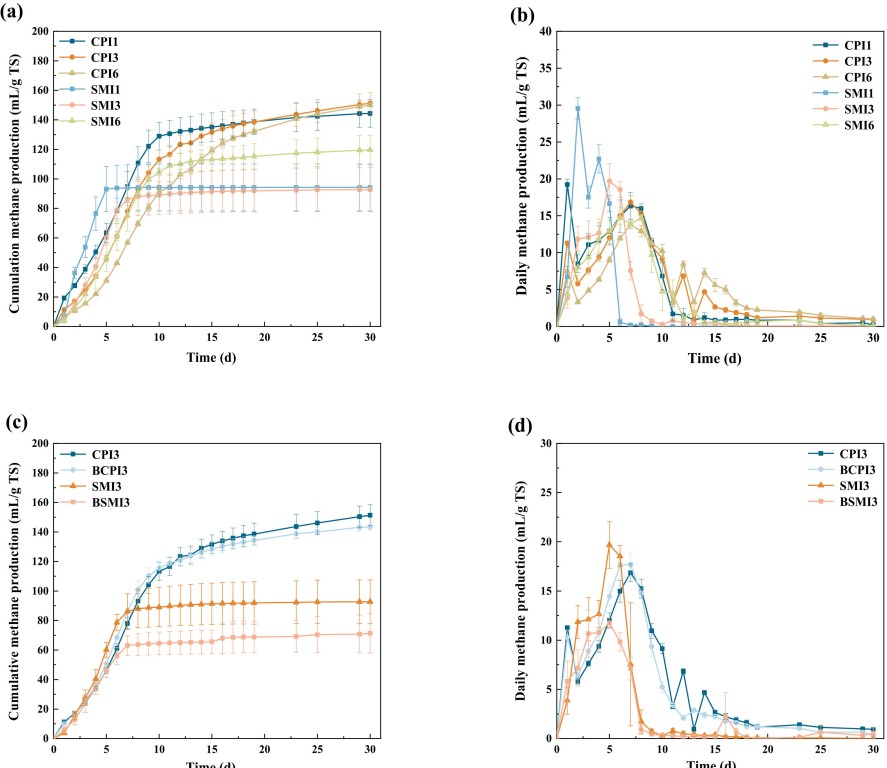

**Figure 2.** Profiles of (**a**) cumulative methane production of different TS concentrations of SM, (**b**) daily methane production of different TS concentrations of SM, (**c**) cumulative methane production of adding biochar to CPI and SMI, (**d**) daily methane production of adding biochar to CPI and SMI.

### 3.1.3. Variations in VFAs and pH in AD

VFAs are the primary intermediate products during the hydrolysis and acidogenesis stages of AD [34], which include acetic acid and propionic acid, and the balance between the output and consumption of VFAs is crucial for the stability of the anaerobic digestion system. Figure 3a shows the change in VFA production during different TS concentrations of swine manure fermentation using different inocula. Without the addition of biochar, the production of VFAs increased with increasing TS concentration of swine manure. The acetic acid accumulation was increased by 45.24–77.5% and propionic acid accumulation was decreased by 21.34–44.41% in the CPI groups compared to the SMI groups. Acetic acid can be directly transformed into $CH_4$ by acetogenic methanogens, while propionic acid is an unfavorable substrate for microorganisms, and ensuring the degradation reaction of propionic acid proceeds spontaneously is difficult [35]. Therefore, the accumulation of propionic acid in VFAs is directly related to the processing capacity of the anaerobic digestion system and such accumulation will lead to a decrease in the methane production of the system [36], which explains why the methane production performance of CPI was better than SMI. Moreover, as shown in Figure 3b, biochar supplementation promoted the production of acetic and propionic acids. Compared with CPI3 and SMI3, the accumulation of acetic acid increased by 6.59% and 12.9%, while the accumulation of propionic acid increased by 30.04% and 76.89% for BCPI3 and BSMI3, respectively, which could be attributed to the surface accumulation or sorption of VFAs onto biochar. A previous study suggested that biochar synthesized at higher temperatures may have a higher absorption capacity due to the higher surface area, as well as their high aromaticity and polarity [37]. The addition of biochar caused more accumulation of propionic acid, which may be one of the reasons for inhibiting methanogenesis. In addition, no propionic acid accumulation was observed in BCPI3 and BSMI3 on day 20, but CPI and SMI still had a slight accumulation of VFAs. It was demonstrated that the addition of biochar can promote the degradation of VFAs, which may be due to the fact that biochar can promote the colonization of microbial that can provide faster metabolism of VFAs [17].

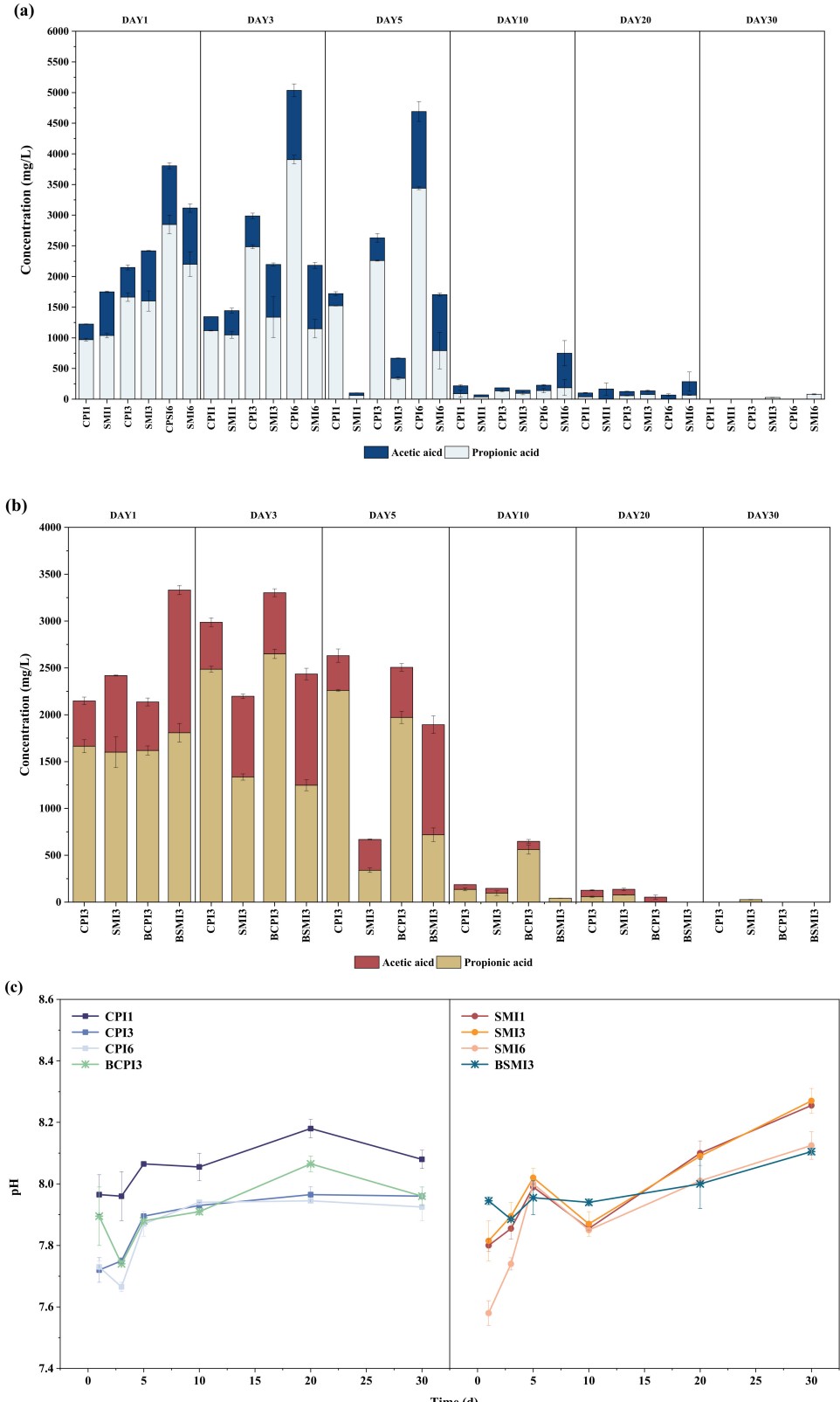

**Figure 3.** Profiles of (**a**) changes in the effect of SMI and CPI on individual VFA in anaerobic di-gestion of swine manure at 1, 3, and 6% TS concentrations, (**b**) changes in the effect of adding biochar on individual VFA in anaerobic digestion of swine manure at 3% TS concentration, (**c**) variation in pH in AD system with different treatment during digestion time.

The pH is an important parameter to monitor metabolic status and process stability during the AD process. In the present study, the pH of CPI groups ranged from 7.72 to 8.18 (Figure 3a), of which the fluctuation range was consistent with the result obtained by Yuan Y [38]. Compared to CPI groups, a larger change in the system pH of SMI groups was observed during the AD process, the pH fluctuated within the range of 7.58 to 8.27, which indicated that the digestion system of CPI groups was more stable than that of SMI groups. However, the addition of biochar to SMI3 (BSMI3) resulted in a decrease in the range of pH fluctuation (7.9–8.1) in the AD system, indicating that the addition of biochar can enhance the stability of the AD system. In addition, the pH value decreased with the increase in organic acids, whereas increased with the increase in ammonia, which was the product of the decomposition of nitrogenous organics [39]. Apparently, the pH of the system for the whole period of 30 days was close to neutral or even slightly alkaline, which also proved that there was a large accumulation of ammonia nitrogen in the AD system. However, a decrease in the pH of BCPI3 and BSMI3 on days 1–3, while not in CPI3 and SMI3. The pH of the experimental group with the addition of biochar (BSMI3) was lower than that of SMI3 on days 15 to 30. The results of the above could further demonstrate that the addition of biochar could promote the production of VFAs and thus alleviate the high ammonia nitrogen stress in the system.

Overall, biochar plays an important role in the improvement of reactor stability through the promotion of VFA production and degradation in the digester.

### 3.2. Microbial Analysis

#### 3.2.1. Diversity and Richness of Microbial Community

The alpha diversity index of the microbial community in four digestate samples was compared, with the results as shown in Table 3. The coverage index for each sample was 100%, indicating that most of the bacterial and archaeal species were detected in these samples. Table 3 shows that 682–814 and 603–745 bacterial operational taxonomic units (OTUs) were detected in the SMI and CPI groups (including SMI, CPI, SMI3, BSMI3, CPI3, and BCPI3), respectively, which were distributed among more than eight phyla. In comparison, the OTUs of archaea were in the range of 35–42, which accounted for a very small proportion of the AD system. ACE and Chao were microbial population estimators that could be utilized to estimate the out number, and their values for bacteria were all higher than those for archaea, indicating high diversity in bacterial communities during digestion. Moreover, the addition of biochar had no significant effect on the variation in OTUs, suggesting that biochar may be mainly influenced by changes in the number of microorganisms, rather than changes in the diversity of the methanogen community.

**Table 3.** The alpha diversity statistics of the microbial community.

| Sample with Days | | Number | | OTUs | | Shannon | | Chao | | Ace | | Shannoneven | | Coverage | |
|---|---|---|---|---|---|---|---|---|---|---|---|---|---|---|---|
| Inoculum | | Bacteria | Archaea | Bacteria | Archaea | Bacteria | Archaea | Bacteria | Archaea | Bacteria | Archaea | Bacteria | Archaea | Bacteria | Archaea |
| SMI | | 40,446 | 36,548 | 682 | 40 | 4.16 | 1.16 | 777.41 | 40.33 | 782.32 | 41.03 | 0.64 | 0.31 | 1.00 | 1.00 |
| CPI | | 46,971 | 52,231 | 603 | 41 | 3.75 | 1.26 | 703.90 | 44.33 | 719.70 | 44.06 | 0.58 | 0.34 | 1.00 | 1.00 |
| SMI3 | 1 | 42,284 | 51,691 | 750 | 42 | 4.29 | 1.55 | 865.56 | 42.75 | 862.01 | 43.51 | 0.65 | 0.41 | 1.00 | 1.00 |
| | 5 | 49,505 | 68,489 | 754 | 38 | 4.44 | 1.17 | 873.54 | 41.00 | 867.66 | 45.70 | 0.67 | 0.32 | 1.00 | 1.00 |
| | 10 | 53,921 | 38,351 | 746 | 39 | 4.50 | 1.15 | 842.00 | 41.00 | 837.32 | 42.77 | 0.68 | 0.31 | 1.00 | 1.00 |
| BSMI3 | 1 | 50,825 | 49,236 | 776 | 41 | 4.28 | 1.60 | 862.47 | 44.75 | 859.37 | 52.30 | 0.64 | 0.43 | 1.00 | 1.00 |
| | 5 | 56,134 | 62,736 | 794 | 43 | 4.47 | 1.41 | 864.49 | 46.00 | 870.09 | 48.55 | 0.67 | 0.38 | 1.00 | 1.00 |
| | 10 | 66,887 | 75,712 | 814 | 41 | 4.50 | 1.51 | 922.33 | 44.75 | 907.90 | 47.90 | 0.67 | 0.41 | 1.00 | 1.00 |
| CPI3 | 1 | 47,148 | 65,583 | 656 | 35 | 3.75 | 1.51 | 793.47 | 45.00 | 771.41 | 40.06 | 0.58 | 0.42 | 1.00 | 1.00 |
| | 5 | 44,174 | 58,738 | 677 | 39 | 4.10 | 1.10 | 785.01 | 39.75 | 799.18 | 41.55 | 0.63 | 0.30 | 1.00 | 1.00 |
| | 10 | 55,700 | 69,863 | 703 | 37 | 4.08 | 0.87 | 814.06 | 38.50 | 816.51 | 38.30 | 0.62 | 0.24 | 1.00 | 1.00 |
| BCPI3 | 1 | 54,292 | 62,894 | 615 | 38 | 3.95 | 1.39 | 706.47 | 38.20 | 684.98 | 39.29 | 0.62 | 0.38 | 1.00 | 1.00 |
| | 5 | 47,809 | 44,592 | 745 | 35 | 4.04 | 1.34 | 889.90 | 35.50 | 869.21 | 35.74 | 0.61 | 0.38 | 1.00 | 1.00 |
| | 10 | 47,022 | 59,001 | 714 | 37 | 4.23 | 1.03 | 825.78 | 38.50 | 828.18 | 40.36 | 0.64 | 0.29 | 1.00 | 1.00 |

#### 3.2.2. Changes in Bacterial Communities

The variation in the relative abundance of the bacterial communities at the phylum level is shown in Figure 4a. Different treatments of anaerobic fermentation approaches exerted different effects on bacterial community composition and diversity. In this study, irrespective of the supplementation of biochar, *Firmicutes*, *Bacteroidetes*, *Synergistetes*, and *Proteobacteria* were the major phylum in each group, accounting for 74.64–84.21% relative abundance (S1).

*Firmicutes*, *Bacteroidetes*, *Synergistetes*, and *Proteobacteria* are typical microorganisms in the anaerobic digestion of livestock and poultry manure, and the major bacterial composition was similar to the one observed in a previous study conducted by Tang et al. [40].

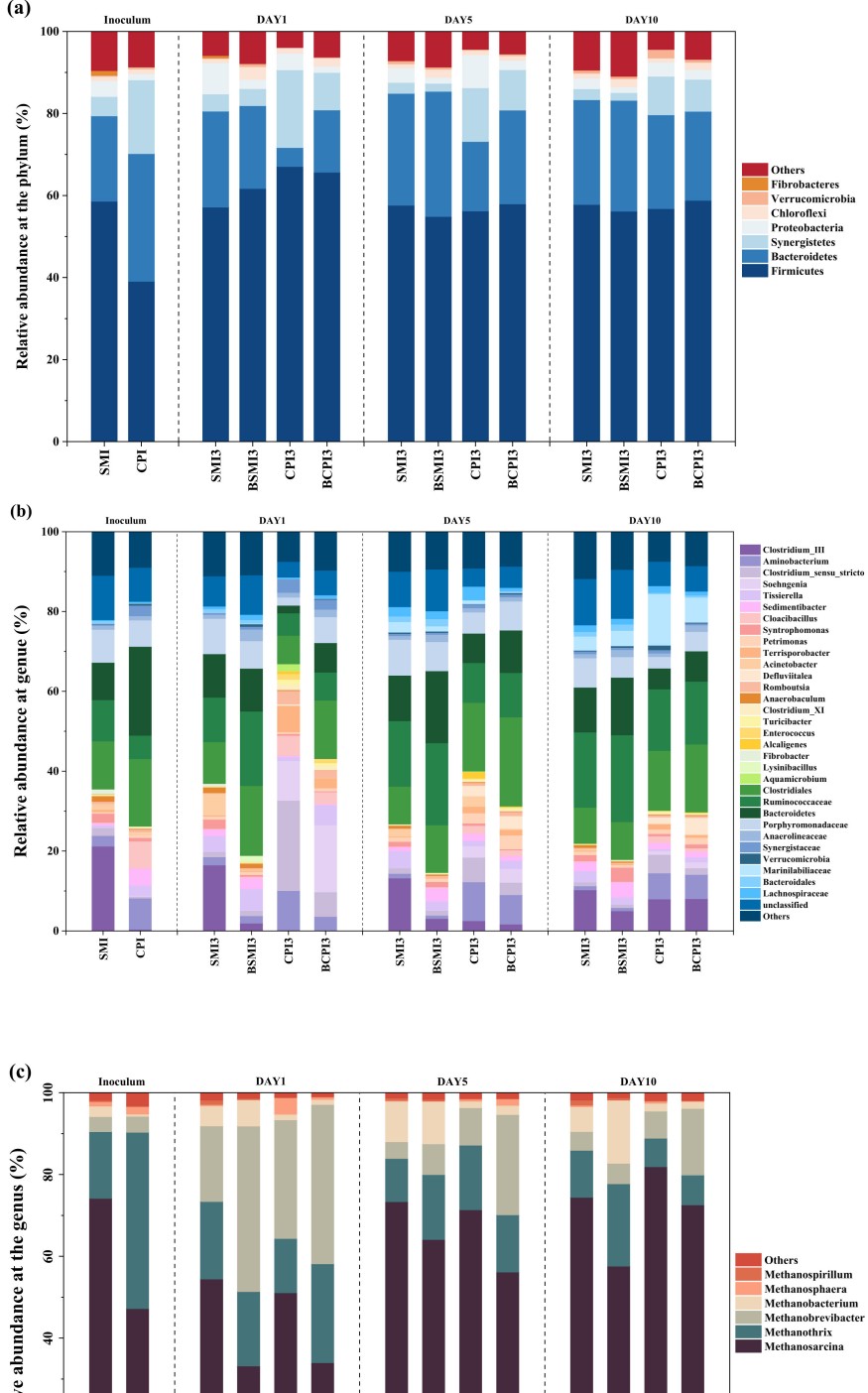

**Figure 4.** Microbial community analysis. (**a**) Relative abundance of bacterial communities at phylum level. (**b**) Relative abundance of bacterial communities at genus level. (**c**) Relative abundance of archaeal communities at genus levels.

The phylum *Firmicutes* accounted for 57.8, 61.7, 67.0, and 65.7% of samples SMI3, BSMI3, CPI3, and CPI3, respectively, and was the most prevalent group among the bacteria identified in all samples. *Firmicutes* are well reported in the literature related to AD, as they are a fundamental step in the realization of a digestion process capable of degrading substrates for the production of volatile acetic acid, $CO_2$, and hydrogen [41]. The phylum *Bacteroidetes* was the second most abundant phylum in the bacterial domain and a type of proteolytic bacteria responsible for the degradation of cellulose and various kinds of proteins to VFAs, succinate, and $NH_3$ [42]. In the present study, the relative abundance of *Bacteroidetes* increased from day 1 to day 5 in all reactors; however, compared to SMI3 (27.25%) and CPI3 (16.93%), the abundance of *Bacteroidetes* in BSMI3 (30.55%) and BCPI3 (22.85%) had increased 12.11% and 34.97%. It indicated that the addition of biochar can promote the growth of proteolytic bacteria and enhance the decomposition of organic matter into VFAs, succinate, and $NH_3$, which corresponded to the result that large amounts of ammonia nitrogen and VFAs were generated in BCPI3 and BSMI3. In contrast, the *Synergistetes* relative abundance in CPI3 and BCPI3 gradually declined; however, in SMI3 and BSMI3, it dropped to a minimum. The abundance of *Synergistetes* increased with the abundance of *Firmicutes* because they are both acetogenic bacteria; however, the abundance of *Bacteroidetes* decreased with increased abundance of them. Therefore, *Synergistetes* was possibly sensitive to environmental changes, including dramatic increases in VFAs and salinity, which are responsible for the decrease in its abundance [43]. Notably, *Synergistetes* is an electrochemically active bacterium involved in DIET, establishing syntrophic metabolism with hydrogen-utilizing methanogens [44], and can ferment amino acids to acetates and, hence, increase biogas production [45]. The relative abundance of *Synergistetes* in BCPI3 and BSMI3 were all decreased compared with CPI3 and SMI3, which corresponds to the phenomenon that the addition of biochar in AD reduces methane production but enhances protein catabolism.

The structure and distribution of the bacterial communities at the genus level are shown in Figure 4b. The results showed that the genera *Clostridium_III*, *Aminobacterium*, *Clostridium_sensu_stricto*, *unclassified_Clostridiales*, *unclassified_Ruminococcaceae*, *unclassified_Bacteroidetes*, *unclassified_Porphyromonadaceae*, and *unclassified_Marinilabiliaceae* were the dominant bacteria genus groups in all reactors throughout the AD process. In particular, the relative abundances of *Clostridium_III* and *unclassified_Ruminococcaceae* were 21.19% and 10.33% in SMI, respectively, and were richer than those in CPI. *Clostridium_III*, a cellulose hydrolytic acidification bacterium with butyric acid as its metabolite [46], was more prevalent in unacclimated digesters; however, its relative abundance remained below 5% in most acclimated digesters [47]. Meanwhile, *Ruminococcaceae* can hydrolyze polysaccharides via different mechanisms, such as producing cellulolytic enzymes, short-chain fatty acids, VFAs, and fermenting hexoses and pentoses [48]. This result further demonstrates that SMI is advantageous for cellulose and polysaccharide decomposition. Furthermore, *Clostridium_sensu_stricto* is a strictly anaerobic bacterium that can readily metabolize various organic substances. *Anaerococcus* can utilize peptone or amino acids as a major energy source to produce VFAs, thus providing supplementary feedstock for methanogenesis. The relative abundance of *Clostridium_sensu_stricto* in CPI3 was 22.62%, consistently higher than that in SMI3 throughout AD, which may explain why CPI had higher methane production than SMI. Moreover, the *Aminobacterium* from a class of *Synergistia* is a kind of proteolytic bacteria, which could degrade many kinds of amino acid produced in AD system [49] and obtained relative abundances of 6.11–9.86% and 0.99–2.65% in CPI and SMI, respectively. This demonstrates that the swine manure inoculum was better at degrading proteins. In addition, the highest abundances in SMI3, BSMI3, CPI3 and BCPI3 were 10.45%, 17.52%, 17.17%, and 22.33%, respectively. It demonstrated that the addition of biochar resulted in a higher abundance of *Clostridiales*, which are carbohydrate and amino acid fermenting bacteria containing genes for carbohydrate-active enzymes that produce propionic and acetic acids [50]. *Tissierella* is the producer of acetic acid [51]. The highest relative abundance of *Tissierella* in SMI3 and CPI3 were 4.24% and 1.31%, which were 20.45% and 73.43%

lower than that in BSMI3 and BCPI3. Hence, this explains why the addition of biochar can improve the degradation of organic substances and enhance the accumulation of VFAs during AD.

Overall, the results demonstrated that SMI was better at degrading cellulose, whereas CPI was better at decomposing proteins as SMI contains more cellulose hydrolytic acidification bacteria, such as *Clostridium_III* and *Ruminococcaceae*, which contain more proteolytic bacteria, including *Clostridium_sensu_stricto* and *Aminobacterium*. Moreover, the addition of biochar to the AD system promoted the growth of key bacteria, such as *Bacteroidetes*, *Clostridiales*, and *Tissierella*, which was in accordance with the high organic substance decomposition rate and removal of SCOD in BCPI3 and BSMI3.

3.2.3. Changes in Archaeal Communities

In the archaeal community, *Methanosarcina*, *Methanothrix*, *Methanobrevibacter*, and *Methanosphaera* were predominant methanogens (Figure 4c). The relative abundances of *Methanobrevibacter* and *Methanobacterium* were higher in BSMI3 and BCPI3 than those in SMI3 and SMI3. In particular, the addition of biochar increased the relative abundance of *Methanobrevibacter* to 40.46% in BSMI3 and to 38.95% in BCPI3, corresponding to improvements of 119% and 34.17% on day 1, respectively, which could be due to *Methanobrevibacter* and *Methanobacterium* attaching to the surface of biochar, and biochar can be provided with nutrients and habitat to promote their growth and reproduction during AD. *Methanobrevibacter* uses $H_2$ and formate as substrates and hydrogenotrophic methanogen [52], which could enhance hydrogenotrophic processes and create a favorable environment for the growth of acetoclastic methanogens [17], such as *Methanothrix*, suggesting that the addition of biochar could improve the stability of the AD system. Despite the fact that *Methanothrix* seemed highly prevalent during the early stages, its relative abundance decreased during AD. In contrast, the relative abundance of *Methanosarcina* in all reactors increased during the AD stage. A possible reason could be that *Methanosarcina* is a hydrogenotrophic methanogen that not only has multiple methanogenic pathways (hydrogenotrophic, acetoclastic, and DIET paths) to produce methane compared with other methanogens but also has a remarkable adaptation ability to compete with other specialized methanogens [53], which lowers the relative abundance of *Methanothrix*. Another reason for this result is that the increase in the average influent VFA content replaced acetoclastic archaea with hydrogenotrophic archaea owing to the susceptibility of *Methanothrix* to VFAs. Notably, the relative abundance of *Methanosarcina* in CPI3 and SMI3 was higher than that in BCPI3 and BSMI3, respectively, which may be an important reason for the reduction of methane production by the addition of biochar. These results can be explained by the following three mechanisms:

(1) Bacteria can communicate with each other by secreting signaling molecules in different microbial systems, which promotes biofilm formation [54]. Due to the high specific surface area and porosity of biochar, microorganisms can readily adhere and grow to form a biofilm on its surface [55]. However, signaling molecules also can be effectively adsorbed on biochar, which is attributed to properties of hydrophobic action, hydrogen bond, and functional group complexation [56], which may affect the type of microbial communities that attach. It has been reported that biochar can enrich *Bacteroidetes*, *Clostridium*, *Anaerolineaceae*, *Clostridiales*, and *Tissierella* [57–60]. Thus, it was further indicated that the biochar will affect the abundance of the microbial community attributed to the more remarkable ability of adhesion of dominant microbes (*Bacteroidetes*, *Clostridiales*, and *Tissierella*) on the biochar's surface [61], which promoted the hydrolysis of organic matter and accumulation of VFAs.

(2) Biochar has shown a significant ability to improve nitrogen removal and metabolic activity attributed to its surface properties [62] and biochar particle size is closely related to the effect of anammox. It is shown that small-sized biochar significantly promoted nitrogen removal efficiency compared with the bigger biochar [63]. In addition, bacteria can access fine particles much more easily than coarse particles and the concentration of

VFAs is higher in fine biochar treatments [17]. Thus, in this study, the addition of biochar can alleviate ammonia stress in AD systems, possibly due to the small particle size biochar accounts for the majority in the mixed particle size biochar, which promotes the decrease in ammonia nitrogen concentration and the accumulation of VFAs.

(3) More biochar may release excessive toxicants, which inhibit methanogenic archaea metabolism [55]. Although biochar can act as an electron mediator to accelerate electron transfer, it also can act as an activator to induce the generation of extracellular and intracellular reactive oxygen species (ROS) [64]. When exposed to a small amount of persistent free radicals, whose concentration correlates with biochar toxicity, methanogenic archaea will be stimulated to produce ROS [65]. Excessive ROS produced are toxic to cells, inhibiting archaea metabolism, and even killing the celling [66]. Therefore, the reduction of methane production in the addition of biochar in this study may be attributed to the limited specific surface area of biochar, the excessive accumulation of VFAs, and the excessive addition of biochar to produce toxins that are not conducive to methanogenic growth.

### 3.3. Challenges and Prospectives for Application of Biochar in Anaerobic Digestion

The above results of this study indicated that biochar enhanced the stability of the AD system by providing a suitable environment for microbes, selectively succeeding, enriching dominant microbes to promote the degradation of organic matter. However, it should be emphasized that this work just confirmed the technical feasibility of biochar addition and revealed its mechanism. As the porous structure of biochar can be used as a carrier for microorganisms, in this sense, biochar acts as inocula that contains rich microorganisms and can be used for other anaerobic digestion to enhance the anaerobic digestion performance, but the effectiveness in AD of biochar may be influenced by the source of the biomass, pyrolysis temperature, particle size, and dosage should be considered before using biochar into other AD systems, especially the ratio of biochar dosage to biomass and even the substrate concentration, which is essential to finding a balance between the promotion and inhibition of biochar. Additionally, additional factors that may influence the effectiveness of biochar in anaerobic digestion processes, such as feedstock characteristics in AD, reactor design, and long-term stability also should be taken into consideration. Moreover, current studies on biochar are still virtually only being conducted in the laboratory context, while large-scale anaerobic digestion is mostly used in practical applications. Therefore, large-scale anaerobic digestion verification tests should be conducted to provide references for practical application.

### 4. Conclusions

The effect of swine manure with different inocula and the addition of biochar to the AD was investigated. The methane production performance of enriched cellulose-peptone-swine inoculant was 20.26–38.71% higher than that of non-enriched swine manure inoculant, whereas the addition of coconut shell biochar not only alleviates ammonia inhibition in AD of swine manure, but also promotes the accumulation of organic acids and the degradation of SCOD, thus improving the stability of AD. Microbial community analysis showed that swine manure inoculum was better at degrading cellulose, whereas cellulose-peptone-swine inoculum was better at decomposing protein. The supplement of biochar elevated the relative abundance of *Bacteroidetes* to enhance proteolysis in AD, and the enrichment of *Methanobrevibacter* and *Methanobacterium* created a stable and favorable growth environment for environmentally sensitive hydrogenotrophic methanogens and improved the performance of AD.

**Author Contributions:** Conceptualization, J.D.; Funding Acquisition, Y.Z. and L.G.; Investigation, J.D.; Methodology, J.D.; Project Administration, F.Z., X.K. and Y.H.; Software, J.D.; Supervision, Y.Z. and L.G.; Validation, Y.Z. and L.G.; Writing—Original Draft, J.D.; Writing—Review and Editing, J.D., F.Z., X.K., Y.H., Y.Z. and L.G. All authors have read and agreed to the published version of the manuscript.

**Funding:** This research was supported by the Technological Project of Heilongjiang Province "the open competition mechanism to select the best candidates" (2022ZXJ05C01), a key R&D project of Heilongjiang Province (GY2021ZB0253/GA21D009), Guangzhou Science and Technology Plan Project (2023B03J1229), China Agriculture Research System of MOF and MARA (cars-35), Guangdong Basic and Applied Basic Research Foundation (2022A1515110188), Science and Technology Program of Guangzhou (202201010683), Start-up Research Project of Maoming Laboratory (2021TDQD002), and Shanwei Science and Technology Plan Project (2023B005).

**Institutional Review Board Statement:** This article does not contain any studies with human participants or animals performed by any of the authors.

**Informed Consent Statement:** Not applicable.

**Data Availability Statement:** All sequence data were uploaded to the NCBI under the study accession number PRJNA1057160.

**Acknowledgments:** The authors would like to thank the support of College of Veterinary Medicine, South China Agricultural University and Guangzhou Institute of Energy Conversion, Chinese Academy of Sciences.

**Conflicts of Interest:** The author Yi Zhang was empoyed by the company Kingfa Sci. & Tech. Co., Ltd. Other authors declare no conflicts of interest.

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
