# Peer review of "Effect of Biochar in Modulating Anaerobic Digestion Performance and Microbial Structure Community of Different Inoculum Sources"

_fermentation, doi:10.3390/fermentation10030151_

Round 1

Reviewer 1 Report (New Reviewer)

Comments and Suggestions for Authors

Author Response

Reviewer 2 Report (New Reviewer)

Comments and Suggestions for Authors

The article is written good but still can be improved.

·       Provide detailed information on the anaerobic digestion setup, biochar characteristics, inoculum sources, and analytical techniques used to assess performance and microbial composition.

·       Provide comprehensive information about the characteristics of the biochar used in the study, including its source, production method, surface area, porosity, and chemical composition. Explain how these characteristics may influence its effectiveness in modulating anaerobic digestion performance and microbial community structure.

·       Discuss the rationale behind choosing specific parameters and experimental conditions, such as temperature, pH, and retention time.

·       Describe the statistical methods used to analyze the data and evaluate the impact of biochar on anaerobic digestion performance and microbial community structure.

·       Interpret the findings in the context of previous research and discuss potential mechanisms underlying the observed effects.

·       Highlight any potential practical applications or considerations for implementing biochar-enhanced anaerobic digestion systems.

·       Consider exploring additional factors that may influence the effectiveness of biochar in anaerobic digestion processes, such as feedstock characteristics, reactor design, and long-term stability.

By addressing these points, the article will be ready for publication.

Author Response

This manuscript is a resubmission of an earlier submission. The following is a list of the peer review reports and author responses from that submission.

Round 1

Reviewer 1 Report

Comments and Suggestions for Authors

The manuscript "Effect of biochar in modulating anaerobic digestion performance and microbial structure community of different inoculum sources" by Jingran Ding, Feng Zhen, Xiaoying Kong, Yunzi Hu, Yi Zhang, and Lang Gong is devoted to study the effect of biochar addition on anaerobic digestion efficiency, biogas production and on changes in prokaryote communities considering the use of different inoculum sources.

However, after a careful reading and judgment, I think this manuscript has some major problems that need to be reviewed:

1.      The composition of the laboratory inoculum (CMI) is not reported. These data should be included in the manuscript.

2.      Table 1.  - What does the abbreviation VS stand for? All other abbreviations are explained in the text of the manuscript.

3.      Section 2.3. should be more detailed.

4.      Section 2.4. – Authors should describe how the bioinformatics processing of the data was performed. Which database was used for identification?

5.      L. 137-138 – Misuse of the term "amplicon" in this context.

6.      L. 171 – How many microbial cells can be adsorbed per unit area of biochar?

7.      L. 184-185 "… which allows for electron transfer via the conductance of carbon matrices and promoted DIET and efficient electron transfer" - Where are the data, research methods on this phenomenon, or references to make this claim?

8.      Fig.1 and L. 173 - Is it reasonable for authors to claim degradation of organic matter in their manuscript? Biochar adsorbs it, so it is difficult to estimate the amount of adsorbed organic matter. This requires clarification.

9.      It is not clear from the text of the manuscript how cumulative methane yield differs from daily methane production?

10.  The authors should explain why the bacterial communities were analyzed only on days 1, 5, and 10, but not after 30 days of the experiment.

11.  Why did the authors use only CMI, SMI, CMI3, BCMI3, BCMI3, SMI3, and BSMI3 samples to analyze the prokaryotic communities and not the others?

12.  Table 4. – "bacteria"  should be replaced by "Bacteria"

13.  L. 289 – "Bacterdetes" should be replaced by "Bacteroidetes"

14.  L. 288-289. Based on Figure 4, two phyla were the most represented with the exception of sample SMI3-5d. An Excel table showing the relative representation of the prokaryotic communities studied at least at the phylum and genus level (since only these taxonomic levels are shown in the figure) should be included in the supplementary materials.

15.  L. 294-295. The word "communities" is missing from the terms "bacterial" and "archaeal".

16.  Raw data from microbial community analysis should be deposited in the NCBI database.

17.  The addition of biochar decreased methane yield and increased the degradation rate of soluble COD organic contamination for different inoculum sources. Therefore, the authors should further discuss in Section 3.3 when biochar should be added and when it should not be added.

Comments on the Quality of English Language

Moderate editing of English language required

Reviewer 2 Report

Comments and Suggestions for Authors

The abstract started with the sentence that using biochar is controversial for some researchers, but you didn't explain why. Please, rewrite the Abstract - now it's a collection of separated sentences rather than the story with the flow. 

In the graphical abstract, you draw that you used cow manure, but there is no information in materials and methods about that.

Line 26: content 

Line 40-41: rewrite and make longer, more complex sentence

The Introduction also does not explain why biochar is controversial. Please include the proper citation. 

Materials and Methods: 

There is no information on how you defined laboratory inoculum - this is a very broad name.

Please specify why you are using cellulose and bacterial peptone, why this particular dynamic of feeding?

What is the meaning of symbols in Table 1? What FM, TS, and VS mean?

Why did you choose 1, 3, and 6 total solid concentrations? Do you have any citations for that? 

What was the pH of the system? Did you adjust that during the 30d experiment?

How did you confirm that anaerobic conditions were achieved?

Line 120-121: please cite those methods, your entire paper is about that.

Checking EC only at days 1 and 30 can be very misleading because you don't have information on what is happening in between. 

In the material and methods of electrical conductivity you mentioned as the last factor and you are starting Results from that, there is no logic in that.

Why in your opinion despite that one of the AD steps is acidogenesis and acetic as well as propionic acids have produced the pH of the system for the whole period of 30 days was close to neutral or even slightly alkaline? In my opinion is very unlikely to maintain such a high pH, since bacterial metabolism usually leads to acidification. You described many results in this paragraph, but the interpretation is missing.

Before you are describing what is the composition of bacterial community in your experiment you have to described what is typical/atypical composition of that in similar environments. 

There are no positive and negative correlations drawn by you between different groups of bacteria that were present/absent in your community. 

Bacterdetes - I believe that should be Bacteroidetes

Moreover, the genera Aminobacterium from class Synergistia, a kind of proteolytic bacteria - too colloquial as for scientific text

In addition, irrespective of CMI3 or SMI3, the addition of biochar-attached biomass resulted in more sequences within Clostridiales, which are carbohydrate-and amino acid-fermenting bacteria containing genes for carbohydrate-active enzymes that produce propionic and acetic acids - you didn't perform any analysis regarding genes presence this conclusion is not justified.

Line 359 and 363: a typo in Mehtanobacterium

The relative abundances of 358 Methanobrevibacter and Mehtanobacterium were higher in BCMI3 and BSMI3 than those in 359 CMI3 and CMI3 - a typo here?

In particular, the addition of biochar increased the relative abundance of 360 Methanobrevibacter to 40.46% in BCMI3 and to 38.95% in BSMI3, corresponding to im- 361 provements of 119% and 34.17% on day 1, respectively, which could be due to Methano- 362 brevibacter and Mehtanobacterium attaching to the surface of biochar, and biochar can pro- 363 vided nutrients and habitat to promote them de-growth and reproduction during AD. 364 - improvements of what? What do you mean by saying "de-growth"? What kind of substrates can be provided by biochar?

There are no Author contributions. 

Comments on the Quality of English Language

I have no comments.

Reviewer 3 Report

Comments and Suggestions for Authors

The manuscript describes the effect of biochar supplementation over manure anaerobic digestion process, experiments include the use of bacterial of laboratory bacterial inoculum and swine manure inoculum at 1, 3, and 6% TS, and two additional biochar (3%) supplemented treatments. Biochar presence reduces methane generation in the systems, but reduces ammonia, increases volatile organic acids generation (acetic and propionic acids), and reduces soluble COD. The presence of biochar induces changes in the microbial community structures, in specific Bacteroidetes bacterial phylum, Methanobrevibacter, and Methanobacterium archaea genus.

The manuscript is well redacted and the topic is interesting, some important format and methodological aspects must be addressed.

Commentaries:

In line 20, “1% to 6%” must be “1 to 6%”

In line 21, “20.26%-38.71%” must be “20.3-38.7%”

In line 22, “5.27% and 23.08%”  must be “5.3 and 23.1%”

In line 32, The graphical abstract includes cow manure in the experiments, but just swine manure was employed, adapt the graphical abstract according to the real experiments carried out.

In line 45, eliminate extra space between word and reference in “manure 3”

In line 62, eliminate extra space between word and reference in “Jo et al. 9”

In line 75, eliminate extra space between word and reference in “Lü et al. 14”

In line 75, eliminate extra space between word and reference in

In line 83, eliminate extra space between word and reference in “10 g/L 17”

In line 85, eliminate extra space between word and reference in “production 18”

In line 101, in “Within days 4” must be “Within day 4”

In line 108, Table 1, define the acronyms “TS”, FM, and VS” the first time used in the manuscript

In line 113, “1%, 3%, and 6%” must be “1, 3, and 6%”

In lines 113-116, Why biochar supplemented experiments with 1 and 6% CMI and SMI were no carried out?

In lines 118-119, How the treatment time was defined in 30 d?, please explain

In Line 121, eliminate extra space between word and reference in “methods 19”

In line 123, “pH 4.00, 7.00, 10.01” could be “pH 4.0, 7.0, 10.0”

In line 154, eliminate extra space in “NH4+-N.  Ammonia”

In line 155, add a space between “2300mg/L”

In line 170, eliminate extra space between word and reference in “SCOD 20”

In line 171, eliminate extra space between word and reference in “cells 21”

In figure 1, panel (a), add a space in “NH4+-N(mg/L)

In lines 183-185, it is mentioned that EC was higher in biochar supplemented treatments, but values are similar, statistical analyses are needed to establish significant differences in EC among treatments

In line 189, eliminate extra space between word and reference in “AD 24”

In line 190, eliminate extra space in “nitrogen  concentration”

In line 195, Table 2 add results of the statistical analyses

In line 202, “34.71%, 38.71%, and 20.26%” must be “34.7, 38.7, and 20.7%”

In line 210, “5.27% and 23.08%” must be “5.3 and 23.1%”

In line 220, eliminate extra space between word and reference in “rate 22”

In figure 2, panels (a), (b) and (c),  add a space between legend and units in the Y axe

In line 233, Table 3, statistical analyses are needed, add results of the table

In figure 3, panels (a), and (b) add a space between legend and units in the Y axe

In figure 3, panels (a), and (b) eliminate the number of days in each indicator at X axe

In line 247, review figure legend, include more information to improve description of the figure 3

In lines 251-255, biochar addition has a beneficial effect on VFA generation, but condition with 6% TS has higher VFA generation, why such condition not was evaluated in biochar supplementation?

In lines 257-259,  propionic acid excessive accumulation reduces methane production, but condition with 1% TS has the lowest propionic acid generation, why such condition not was evaluated in biochar supplementation?

In line 282, eliminate extra space in “digestion.  Moreover”

In lines 282-284, review redaction it is not clear, check the redaction, how biochar is influences by the number od microorganisms or the changes in the diversity of the methanogen community?

Check the format of the all-column legends in table 4

In figure 4, panels (a), (b) and (c),  add a space between legend and the percentage symbol

In line 296, “57.77%, 61.68%, 67.04%, and 65.65%” must be “57.8, 61.7, 67.0, and 65.7%”

In line 300, check the format in “hydrogen. 32.”

In line 329, eliminate extra space in “enzymes,  short-“

In line 394, in the section “3.3. Prospects for Further Application of Biochar” add information about suitability of the use of biochar for the treatment of livestock manure

In line 421, eliminate extra space between word and reference in “crop yield 48” and “Wu et al. 49found”

In lines 437-440, complement conclusions with information about suitability of the use of biochar for the treatment of livestock manure

Round 2

Reviewer 1 Report

Comments and Suggestions for Authors

Overall comments.

 Section 2.4. - Why did the authors use the UNITE fungal ITS database if they did not analyse the fungal communities?

"We have deposited raw data in the NCBI database. (BioProject ID: PRJNA1057160)". - This should be stated in the text of the manuscript.

Figure 1 - What does "Other" mean? This should be stated in the figure caption.

L. 319-322 - According to Figure 4 and supplementary material (S1), only Firmicutes and Bacteroidetes were the dominant phyla.

Reviewer 3 Report

Comments and Suggestions for Authors

The authors addressed adequately all reviewer comments, I consider the manuscript could be suitable for publication in Agronomy Journal.
